# A Large-Scale Dataset of Conservation and Deep Tillage in Mollisols, Northeast Plain, China

**Fahui Jiang** [1,*] **, Shangshu Huang** [1,2] **, Yan Wu** [2] **, Mahbub Ul Islam** [3] **, Fangjin Dong** [4] **, Zhen Cao** [5] **, Guohui Chen** [6] **and Yuming Guo** [7]

1   College of Land Resources and Environment, Jiangxi Agricultural University, Nanchang 330045, China
2   Jiangxi Institute of Red Soil, Nanchang 330046, China
3   Bangladesh Agricultural Research Institute, Gazipur 1701, Bangladesh
4   Northeast Institute of Geography and Agroeocology, Chinese Academy of Sciences, Harbin 150081, China
5   Nanjing Institute of Geography and Limnology, Chinese Academy of Sciences, Nanjing 210008, China
6   Jiujiang Academy of Agricultural Sciences, Jiujiang 332000, China
7   Cultivated Land Quality Monitoring and Protection Center, Ministry of Agriculture and Rural Affairs, Beijing 100125, China
*   Correspondence: fhjiang@issas.ac.cn; Tel.: +86-183-6208-6809

**Abstract:** One of the primary challenges of our time is to feed a growing and more demanding world population with degraded soil environments under more variable and extreme climate conditions. Conservation tillage (CS) and deep tillage (DT) have received strong international support to help address these challenges but are less used in major global food production in China. Hence, we conducted a large-scale literature search of English and Chinese publications to synthesize the current scientific evidence to evaluate the effects of CS and DT on soil protection and yield maintenance in the Northeast China Plain, which has the most fertile black soil (Mollisols) and is the main agricultural production area of China. As a result, we found that CS had higher soil bulk density, strong soil penetration resistance, greater water contents, and lower soil temperature, and was well-suited for dry and wind erosion-sensitive regions i.e., the southwest areas of the Northeast. Conversely, DT had better performance in the middle belt of the Northeast China Plain, which contained a lower soil temperature and humid areas. Finally, we created an original dataset from papers [dataset 1, including soil physio-chemical parameters, such as soil water, bulk density, organic carbon, sand, silt, clay, pH, total and available nitrogen (N), phosphorus (P), and potassium (K), etc., on crop biomass and yield], by collecting data directly from publications, and two predicted datasets (dataset 2 and dataset 3) of crop yield changes by developing random forest models based on our data.

**Dataset:** https://www.mdpi.com/article/10.3390/data8010006/s1

**Dataset License:** Creative Commons Attribution 4.0 International.

**Keywords:** conservation tillage; deep tillage; conventional tillage; random forest; meta-analysis; subsoiling; no-tillage; straw mulching; crop yield

## 1. Summary

One of the primary challenges in our future is to feed a growing and more demanding world population with degraded soil, a shortage of nutrients, and strong soil erosion caused by water and wind—all under more variable and extreme climate conditions [1–3]. In past decades, soil degradation, such as soil compaction, organic matter, nutrient losses, and reductions in soil-ploughed layer thickness, was observed in conventional (traditional) tillage with the practices of mechanization and intensification, although such intensive tillage has provided plenty of crop yields [4]. Therefore, a new agricultural strategy to

balance the needs of the present and future was an urgent need, especially since such tillage practices provide high yields with fewer negative impacts on the soil.

Conservation tillage aims to minimize or reduce the loss of soil and water that has received strong international support as a challenge that needs to be addressed. With recent conservation tillage, the widespread adoption of no-till has existed over approximately 125 million hectares across the USA, sub-Saharan Africa, and South Asia [5,6]. However, it is less used in the major global food production areas, such as China, due to concerns about the negative effects of no-tillage on crop yield. Conversely, deep tillage was one of the most effective ways to retain or increase crop yield in the field by enhancing the plant availability of subsoil nutrients and water resources, as well as reducing soil compaction [7]. Where the subsoil layer is lower than the normal ploughing layer, it can store almost 50% and 25–70% of total nitrogen and phosphorus stocks, respectively [8,9]. These subsoil resources can provide new nutrients and water to help partly solve the nutrient shortage. Nonetheless, this technology is also underestimated in the cropland of China.

We, therefore, conducted a large-scale literature search of the main grain-producing areas to synthesize the current scientific evidence and analyze the negative and positive effects of conventional tillage or deep tillage on soil properties and crop yield in Mollisols, Northeast Plain, China. Then, we published an article in an influential Chinese journal aimed at creating a better public policy for China's government [10]. However, in the previous study, we focused on the analysis of the data and only briefly described the data acquisition process, due to space constraints.

In this work, we describe our dataset acquisition process in detail and provide it in English to improve international conservation and deep tillage research work. The aims of this paper were to provide an original dataset from papers and two predicted datasets of crop yield changes in differential tillage practices based on random forest models, as well as details of the scientific methodologies of the publications' data collection.

## 2. Data Description

### 2.1. Original Dataset from Papers (Dataset 1)

This article provides crop yield and soil property datasets (in an Excel format) for conservation tillage (CS) compared with conventional tillage (CT), and deep tillage (DT) compared with conventional tillage (CT), in Mollisols of the Northeast China Plain, which was compiled from English and Chinese publications through a literature search. We describe the details of this dataset in the following section:

### 2.1.1. The Basic Information of Each Field Experiment

- ID: the number code of the paper, with each paper receiving one number;
- Title: the title of the paper, with Chinese papers translated into English;
- Author: the name of the paper's first author;
- Journal: the publication journal of the paper;
- Country/Province/City: the location of each experiment;
- Lng(E)/Lat(N): the longitude and latitude coordinates of each experiment's site; the east longitude and north latitude are positive numbers, and the west longitude and south latitude are negative numbers;

### 2.1.2. The Local Climate and Topography of Experiment Sites

- Temperature (°C): annual mean air temperature of each site from the nearest weather station (not the soil layer temperature);
- Precipitation (mm): the local annual mean rainfall data from the nearest weather station;
- Available accumulated temperature (°C): annual mean accumulated temperature, which is the sum of the temperatures > 10 °C;
- Aridity index: the aridity index of the sites, which is the ratio of the annual potential evapotranspiration (ET0) and the annual precipitation. In this paper, we used the average value of 1960–2013 [11];

- Humidity index: the reciprocal of the aridity index;
- Slope ($°$): the slope of the sites;
- DEM (mm): the digital elevation model of the site, which reflects the altitude;

### 2.1.3. The Time of Field Experiment and Sample Collection

- Start and end time: the beginning and ending time of the experiment;
- Duration (yr): the duration, in years, between the start and end;
- Sample time: the time of soil and yield collection;

### 2.1.4. The Initial Soil Physio-Chemical Properties of Experiments

- N/P/K: soil total nitrogen, total phosphorus, and total potassium contents before the experiment, in units of g $kg^{-1}$;
- a.N/a.P/a.K: soil-available nitrogen, available phosphorus, and available potassium concentration before the experiment, in units of mg $kg^{-1}$;
- SOC: soil organic carbon contents before the experiment, in units of g $kg^{-1}$;
- Bulk density: soil bulk density before the experiment, in units of g $cm^{-3}$;
- Sand/Silt/Clay: soil texture of the sites, %;
- pH: soil pH value before the experiment.

### 2.1.5. The Human Management Practices

- Land use: upland soil only;
- N/$P_2O_5$/$K_2O$: the amount of applied N, P, and K fertilizer in each cropping season, in units of kg $ha^{-1}$;
- OM: whether to use manure or organic matter, yes or no;
- Irrigation: whether to irrigate or not, yes or no;
- Crop: the crop in the current season, maize or soybean;
- System: the cropping system, including mono only in Northeast China.

### 2.1.6. The Conservation/Deep/Conventional Tillage Details

- Tillage time: we divided the tillage time into "middle" and "pre-plant" according to the till time. The middle is tillage in the seedling stage and the pre-plant is tillage before seeding, which contains spring and autumn ploughing after the previous crop harvest;
- Treat-tillage: the treatment of tillage, including deep ploughing more than 25 cm, subsoiling more than 25 cm, and no-tillage;
- Tillage-depth: the average depths of the tillage practice if they give one max and one min value, or else using the only value;
- Straw-treat: the previous method of straw retention, such as incorporation with deep tillage, mulching over the ground, or not mentioned by the author (not clear);
- Control-treat: conventional tillage, including rotary tillage of 15–18 cm or autumn ridge tillage near 15 cm;
- Number-treat and number-control: the replication number of the experiment.

### 2.1.7. The Soil Physio-Chemical Properties Changing after Several Years of Tillage

- Physical parameters: include soil bulk density during the mature or seeding time, penetration resistance during the mature time, soil moisture during the seeding, growing and mature periods, soil layer temperature during the seeding, growing and mature periods, soil mac-aggregate, and soil aggregate mean weight diameter (MWD);
- Chemical parameters: soil organic carbon contents (SOC) in differential layers, total nitrogen (TN), total phosphorus (TP), and total potassium (TK) contents in differential layers, etc., and soil-available nitrogen (a.N), available phosphorus (a.P), and available potassium (a.K) concentrations in differential layers; in addition, the soil pH was also collected;

### 2.1.8. The Crop Yield and Biomass after Several Years of Tillage

- Root: the root length or root weight in unit volume soils;
- Crop biomass: the above-ground biomass i.e., the dry matter weight of crop straw, in units of kg ha$^{-1}$;
- Crop yield: the yield of maize or soybeans after harvest, in units of kg ha$^{-1}$.

### 2.1.9. Overview of Dataset Parameters Changed in CS vs. CT and DT vs. CT Systems

- The boxplots (Figure 1) were used to show the data distribution, which included the interquartile range, max, min, median, and outliers of each parameter. Overall, the soil properties and crop-growing parameters were distributed around zero with fewer outlier values, which means the dataset has high data quality.
- Furthermore, CS had higher soil bulk density, strong soil penetration resistance, more water content, lower soil temperature, lower mac-aggregate, a small increase in soil nutrients, higher crop biomass, and fewer changes in crop yield compared with CT. These results were similar to much other research on a global scale [12,13], in Africa [14,15], and regions of the USA [16].
- Conversely, DT had a lower bulk density and penetration resistance, higher soil water and temperature than CT, as well as a larger amount of mac-aggregate and organic carbon matter, with more soil nutrients, under- and above-ground biomass, and higher crop yield gains. This finding is also supported by a number of other reports [7,17].

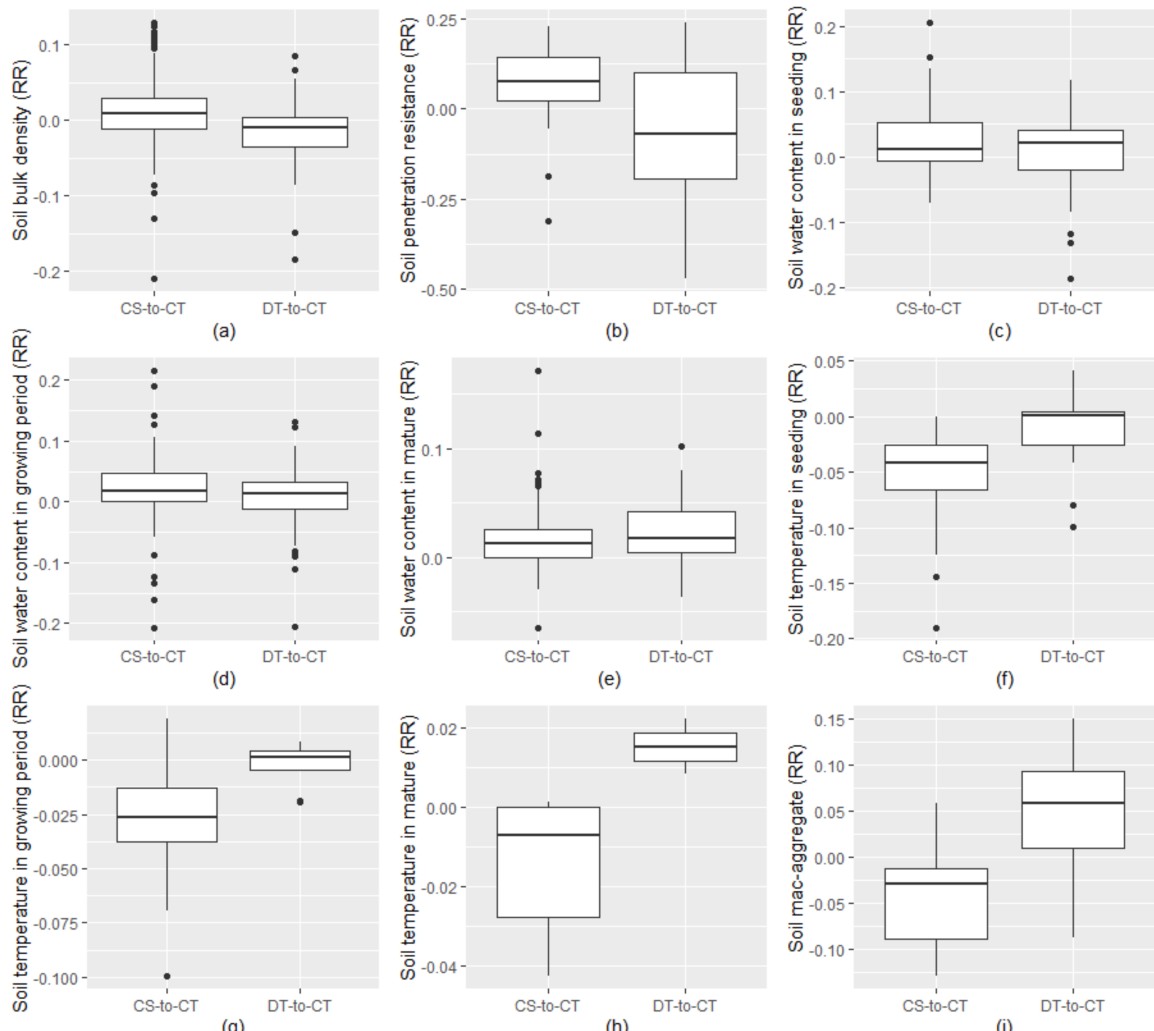

**Figure 1.** *Cont.*

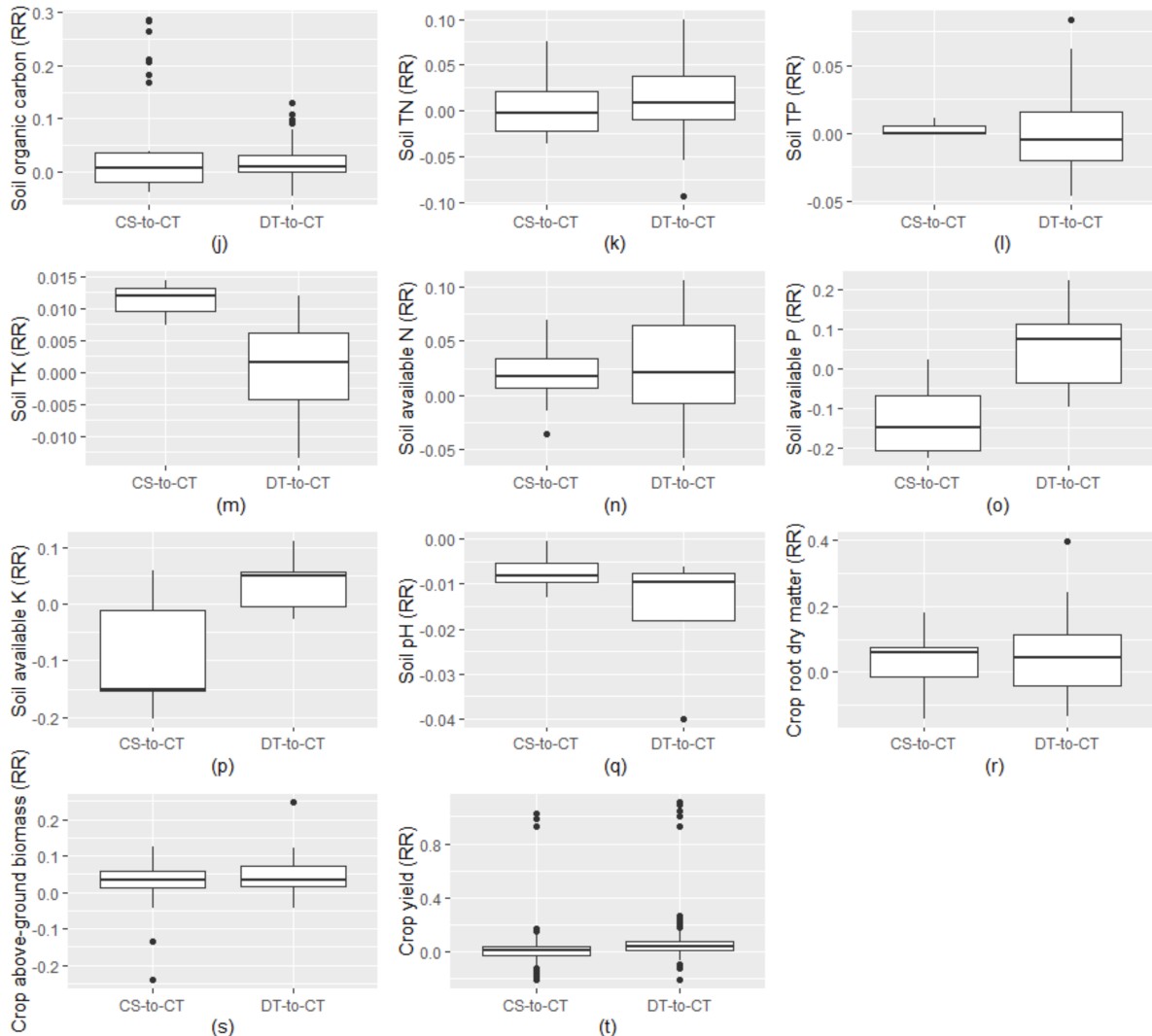

**Figure 1.** The overview of dataset parameters changed under conservation (CS) vs. conventional tillage (CT), and deep (DT) tillage vs. conventional tillage (CT). (**a**) Soil bulk density, (**b**) Soil penetration resistance, (**c**) Soil water content in seeding period, (**d**) Soil water content in growing period, (**e**) Soil water content in mature period, (**f**) Soil temperature in seeding period, (**g**) Soil temperature in growing period, (**h**) Soil temperature in mature period, (**i**) Soil mac-aggregate, (**j**) Soil organic carbon, (**k**) Soil total nitrogen (TN), (**l**) Soil total phosphorus (TP), (**m**) Soil total potassium (TK), (**n**) Soil available nitrogen (N), (**o**) Soil available phosphorus (P), (**p**) Soil available potassium (K), (**q**) Soil pH, (**r**) Crop root dry matter, (**s**) Crop above-ground biomass, and (**t**) crop yield. The boxplot has the min, max, and interquartile range, within which the solid and black dots represent the median and outlier values, respectively.

### 2.2. Predicted Dataset of Crop Yield

In the second part, the large-scale distribution of predicted relative changes in crop yield under CS-to-CT and DT-to-CT was projected onto basic cropland of the Northeast China Plain using a machine learning language (random forest regression model) to incorporate the local topography, climate, initial soil properties, etc., on human management practices data.

#### 2.2.1. Predicted Yields under CS-to-CT (Dataset 2)

An example figure of dataset 2 is shown in Figure 2 (available in tif format to be downloaded at the end). In this figure, the percent changes of the predicted yield response ratios (RR) of CS-to-CT were projected onto the cropland of the Northeast China map,

with a spatial resolution of 1 km$^2$ (more detail provided in Section 3). In this dataset, the average predicted yield under CS-to-CT was 8.11%, with the highest and lowest values being 13.7% and −5.7%, respectively (Figure 2). Overall, the predicted yields in the southwest of China's Northeast Plain (red color) were higher than in the north and east (green color), which indicates that conservation tillage (CS) had a better performance in the southwest. Regions with large productivity gains under CS corresponded well with regions with large dry stress [11] and wind erosion sensitivities [18] in our dataset, which mainly contributed to the straw mulching and no-tillage of CS. This, in turn, increased the soil water content (Figure 1c–e) and reduced the soil temperature (Figure 1f–h) as well as the soil disturbance [19]. Furthermore, we can calculate the areas of CS adoption in Northeast China depending on this dataset and then guide farmers in making better decisions in their agricultural practices.

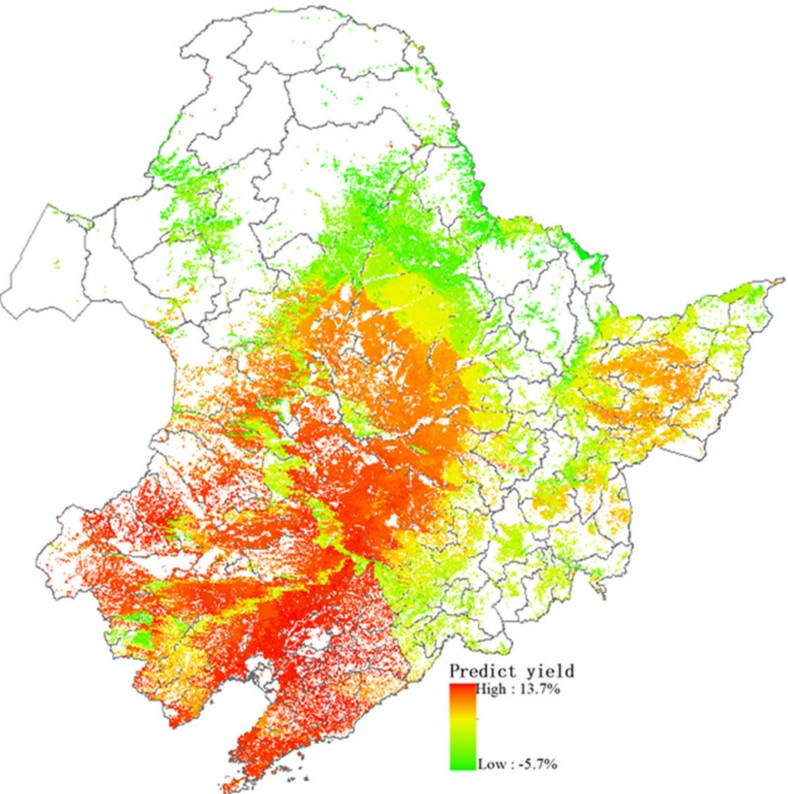

**Figure 2.** An example of dataset 2; the distribution of the predicted yields under CS−to−CT in Mollisols of the Northeast China Plain.

### 2.2.2. Predicted Yields under DT-to-CT (Dataset 3)

An example figure of dataset 3 is shown in Figure 3 (the tif file within the available predicted data is also provided in the Supplementary materials). The results revealed that deep tillage (DT) had a better yield performance in almost the whole Northeast China Plain when compared with traditional tillage (CT). The average predicted yield in the DT-to-CT system was high, 10.34%, across all areas, and the lowest value was approximately zero (which means equal yields of DT and CT) at −1.3%. Indeed, the negative value of the predicted yield was near 0% of the total amount of data (data not shown). Moreover, if we compared the yield performance of the DT system only, the results showed that deep tillage was better in the middle belt of the Northeast China Plain than in other areas (Figure 2). The associated mechanisms can be manifold. Crop seedling emergence is limited by the cold, and growth is influenced by waterlogging in the middle belt, which contained a lower temperature, higher humidity, and low laying [10]. DT performs well to help address these

challenges by increasing soil temperature during the seeding period (Figure 1f) and soil moisture infiltration during the growing season [10].

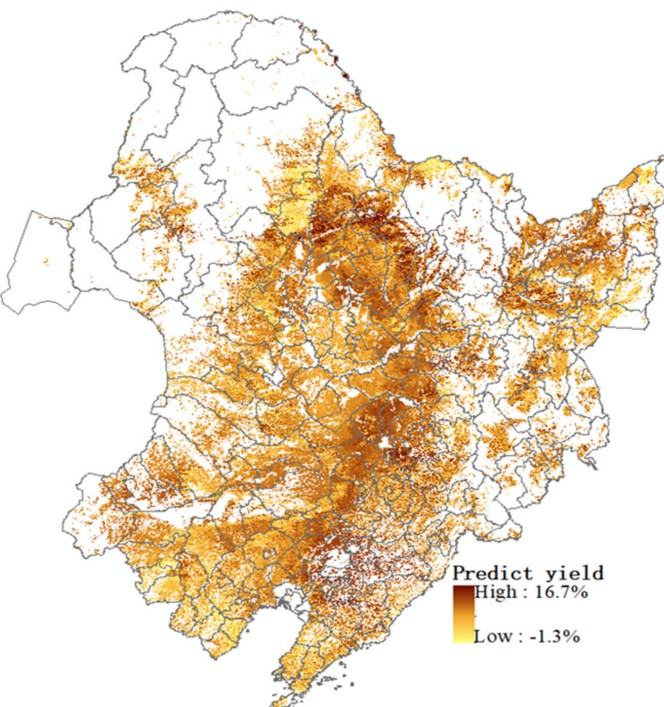

**Figure 3.** An example of dataset 3; the distribution of the predicted yields under DT-to-CT in Mollisols of the Northeast China Plain.

## 3. Methods

### 3.1. Original Dataset from Papers

3.1.1. Literature Search and Data Collection

We conducted a literature search of peer-reviewed scientific journal articles using Web of Science (ISI), Science Direct, Springer, Google Scholar, Scopus, the China Knowledge Resource Integrated (CNKI) database, China's science and technology journal database, and the China Wan-fang database for field-based experiments on the impact of conservation (CS) or deep tillage (DT) on soil properties and crop yield compared with the conventional tillage (CT) (see details in Table 1). In doing so, the main keywords, including conservation/reduce/no-tillage or straw return/mulching or deep tillage/subsoiling/deep ripping/deep mixing/vertical deep rotary, combined with the north/northeast China on black soil or Mollisols, were used to search for relevant papers in English and Chinese until November 2021 with no limitations on the publication year. Then, some inclusion criteria were applied to screen for appropriate articles (see Figure 4).

**Table 1.** The details of tillage types included in the dataset.

| Tillage Type | Tillage Practice | Tillage Depth |
|---|---|---|
| Conservational tillage (CS) | reduce tillage or no-tillage with straw returning | |
| Deep tillage (DT) | subsoiling, deep ploughing, deep mixing, vertical deep rotary tillage | >25 cm |
| Conventional tillage (CT) | rotary, disk plow, shallow inverse, rotary with ridge | <17 cm |

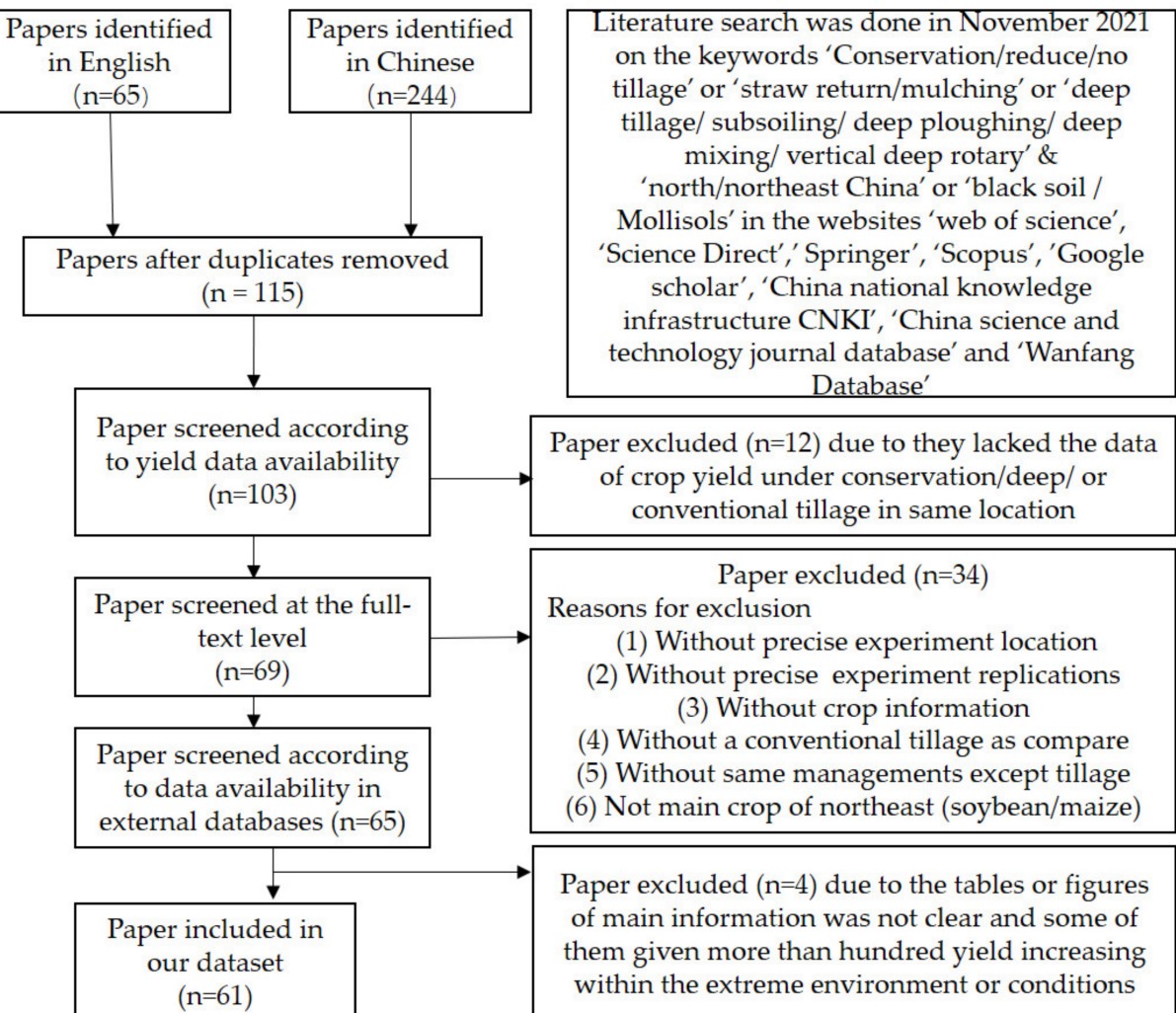

**Figure 4.** Flow chart of literature search and data collection.

Finally, 61 publications (see details in supplementary excel file) fulfilled the specific criteria for inclusion in this dataset, and the original dataset (see details in part 2.1) was collected from these papers, directly from the tables or extracted from figures, using Web Plot Digitizer 4.2 software. According to the meta-analysis protocols, we collected the mean (average value), standard deviation (SD), and the number of CS/DT or CT treatments (if only standard error (SE) was reported, SD was calculated by SD = SE × $\sqrt{n}$) for each parameter of soil properties or crop-growing changes, thus helping to meet the most popular multi-experiment, quantitative scientific synthesis methodology [20]. In addition, basic information from papers or experiments was also collected to explain the reasons for parameter changes.

### 3.1.2. Data Distribution and Quality Assurance

The conservation and deep tillage field experiments were conducted in differential sites in cropland areas and mostly in agriculturally active regions of Northeast China (Figure 5). In total, 71 experiment sites were collected in this dataset, and there were sites distributed across the Liaoning, Jilin, and Heilongjiang Provinces, which cross every agro-climatic zone (including humid, semi-humid, semi-arid, and arid regions of Northeast China). For further analysis of the effects of the climate and region properties on soil or crop yield response to different tillage, we categorized our research sites into three main agricultural plains: the Songnen, Sanjiang, and Liao River Plains (see details in the pre-paper of Jiang, 2021).

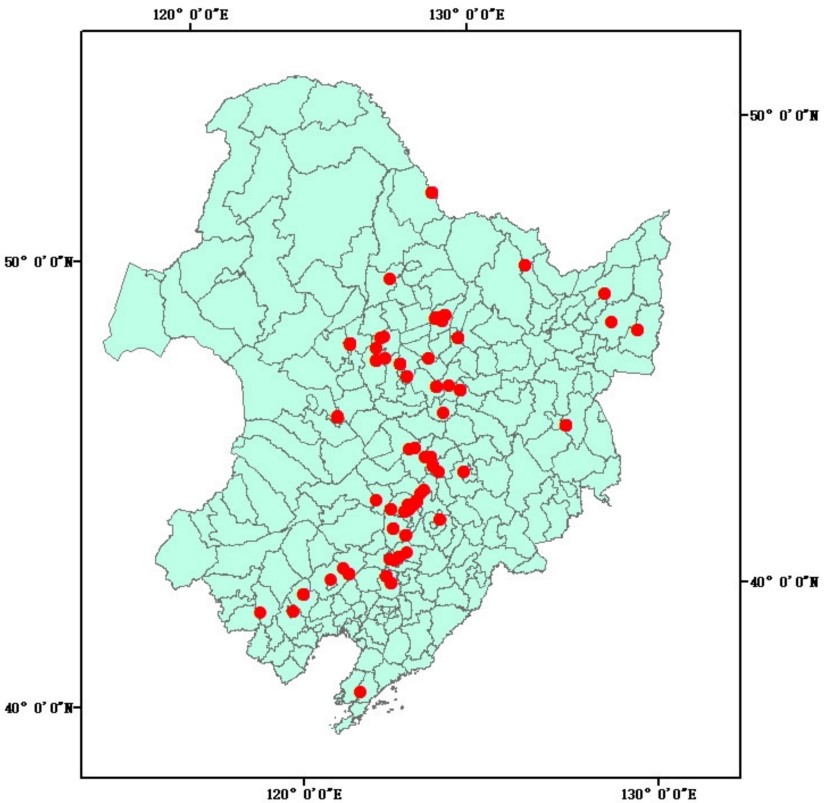

**Figure 5.** Locations of experiments included in the dataset.

There are several data biases that would influence the quality of one dataset collected from many scientific reports, such as publication bias, delay bias, multiple/repeated publication bias, publication position bias, citation bias, language bias, and result-reporting bias (Cochrane Handbook). In this paper, we used the endnote "find duplicates" tool and manually removed any repeated papers or experiments to avoid multiple/repeated publication bias and keep the data independent. We used the fail-safe N technique to test the most impactful potential bias, publication bias, before constructing contour-enhanced funnel plots and quantifying their asymmetry through Egger's test [21]. The test results are shown in Figure 6. Overall, Egger's tests were non-significant for most of our dataset parameters—for example, soil penetration resistance ($p = 0.081$, Z = 1.7), soil water content (seeding: $p = 0.26$, Z = 1.12; growing period: $p = 0.32$, Z = −0.98; mature: $p = 0.43$, Z = −0.78), soil temperature (seeding: $p = 0.18$, Z = 1.33; growing period: $p = 0.23$, Z = 1.21; mature: $p = 0.19$, Z = 1.32), soil mac-aggregate ($p = 0.66$, Z = −0.44), soil total nutrients (TN: $p = 0.32$, Z = −0.98; TP: $p = 0.42$, Z = −0.802; TK: $p = 0.08$, Z = −1.73), soil-available nutrients (a.P: $p = 0.96$, Z = 0.048; a.K: $p = 0.31$, Z = −1.002), soil pH ($p = 0.58$, Z = −0.56), crop root dry matter ($p = 0.78$, Z = −0.28), crop biomass ($p = 0.22$, Z = −1.22), and the crop yield ($p = 0.6$, Z = −10.6). The $p$ values were > 0.05 for all of the above-mentioned indicators; that is, there was no significant publication bias within these parameters. However, some parameters existed with significant publication bias, such as soil bulk density ($p = 0.0024$, Z = −3.03), soil organic carbon ($p = 0.0029$, Z = 2.975), and soil-available nitrogen ($p = 0.0008$, Z = −3.35). Thus, more caution is needed when analyzing and drawing conclusions from these data. In general, this original dataset from papers has high quality assurance.

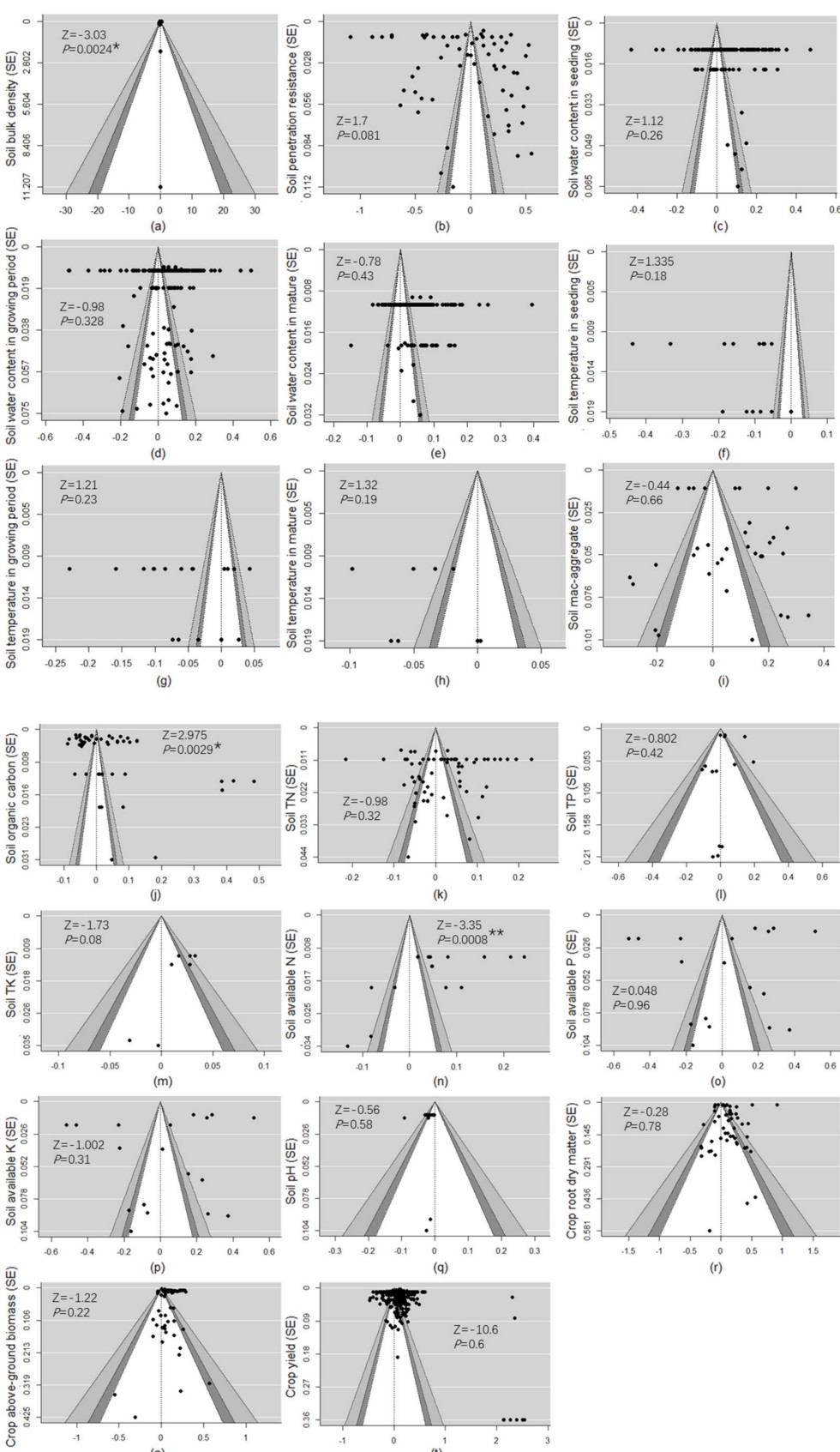

**Figure 6.** Funnel plot of data under conservation (CS) vs. conventional tillage (CT), and deep (DT) tillage vs. conventional tillage (CT). (**a**) Soil bulk density, (**b**) Soil penetration resistance, (**c**) Soil water content in seeding period, (**d**) Soil water content in growing period, (**e**) Soil water content in mature

period, (**f**) Soil tempera-ture in seeding period, (**g**) Soil temperature in growing period, (**h**) Soil temperature in mature period, (**i**) Soil mac-aggregate, (**j**) Soil organic carbon, (**k**) Soil total nitrogen (TN), (**l**) Soil total phosphorus (TP), (**m**) Soil total potassium (TK), (**n**) Soil available nitrogen (N), (**o**) Soil available phosphorus (P), (**p**) Soil available potassium (K), (**q**) Soil pH, (**r**) Crop root dry matter, (**s**) Crop above-ground biomass, and (**t**) crop yield. Egger's test of the intercept (1997) quantifies the funnel plot, and the fail-safe N technique was used to test for publication bias. The $p$ and Z values are shown in the figures, with significant (*) $p < 0.05$ and extreme-significant (**) $p < 0.001$ findings of publication bias indicated.

## *3.2. Predicted Dataset of Crop Yield*
### 3.2.1. Support Datasets Collection

The basic Northeast datasets of climate, local surface topography, soil properties, etc., on some human management practices were collected from various sources to support model settings for cropland projections of the predicted yield gain's relative change under CS-to-CT and DT-to-CT (Table 2). All the datasets from differential resolution were normalized into a spatial resolution of 1 km$^2$ using the "GeoPandas" package from version 37 of the python software.

**Table 2.** The various sources of basic datasets for the Northeast.

| Model Input | Setting (for Each Grid Cell and Each Crop) | Source |
|---|---|---|
| Precipitation | Mean annual precipitation 2006–2015 | Resource and Environment Science and Data Center https://www.resdc.cn/Default.aspx. accessed on 11 March 2022. |
| Temperature | Mean annual temperature 2006–2015 | Resource and Environment Science and Data Center https://www.resdc.cn/Default.aspx. accessed on 11 March 2022. |
| Accumulated temperature | Mean annual accumulated temperature > 10 °C,2006–2015 | Resource and Environment Science and Data Center https://www.resdc.cn/Default.aspx. accessed on 11 March 2022. |
| Aridity index | Mean annual aridity from 1990 until now | Resource and Environment Science and Data Center https://www.resdc.cn/Default.aspx. accessed on 11 March 2022. |
| Humid index | Mean annual humid-index from 1990 until now | Resource and Environment Science and Data Center https://www.resdc.cn/Default.aspx. accessed on 11 March 2022. |
| Digital elevation model (DEM) | Surface 30 m digital elevation model | Geospatial Date Cloud: http://www.gscloud. accessed on 02 March 2022. |
| Site slope | Surface 30 m resolution land slope | Geospatial Date Cloud: http://www.gscloud. accessed on 02 March 2022. |
| Soil bulk density | An average of each grid cell (1 km × 1 km) | Institute of Soil Science, Chinese Academy of Sciences, Nanjing http://doi.org/10.11666/00073.ver1.db. accessed on 13 March 2022. |
| Soil organic carbon | An average of each grid cell (1 km × 1 km) | Institute of Soil Science, Chinese Academy of Sciences, Nanjing http://doi.org/10.11666/00073.ver1.db. accessed on 13 March 2022. |
| Soil total nitrogen | An average of each grid cell (1 km × 1 km) | Institute of Soil Science, Chinese Academy of Sciences, Nanjing http://doi.org/10.11666/00073.ver1.db. accessed on 13 March 2022. |

**Table 2.** *Cont.*

| Model Input | Setting (for Each Grid Cell and Each Crop) | Source |
|---|---|---|
| Soil clay content | An average of each grid cell (1 km × 1 km) | Institute of Soil Science, Chinese Academy of Sciences, Nanjing http://doi.org/10.11666/00073.ver1.db. accessed on 13 March 2022. |
| Soil pH | An average of each grid cell (1 km × 1 km) | Institute of Soil Science, Chinese Academy of Sciences, Nanjing http://doi.org/10.11666/00073.ver1.db. accessed on 13 March 2022. |
| Cropland | The agricultural upland soil areas of each grid cell (1 km × 1 km) | Institute of Soil Science, Chinese Academy of Sciences, Nanjing http://doi.org/10.11666/00073.ver1.db. accessed on 13 March 2022. |
| Irrigation | Yes or no | |
| Soil tillage | CS-to-CT/DT-to-CT | |
| Crops | Maize or soybean | |

### 3.2.2. Multivariate RF Regression Modelling

A random forest (RF) regression model was created to predict the distribution of the relative crop yield under CS-to-CT and DT-to-CT [22]. The basic information, including the local climate, initial soil properties, and human management, was incorporated into the RF model using multivariate statistical regression. Specifically, RF needs were developed first, and then the following procedures were predicted as shown in the following stages.

First of all, the response ratios (RR) of the crop yields to CS- to-CT and DT-to-CT were calculated as RR = ln $(\frac{Yield_{CS/DT}}{Yield_{CT}})$, with the yields of CS: conservation tillage, DT: deep tillage, and CT: conventional tillage. Further, the explanatory variables were selected from the original dataset from papers based on the relative importance of the factors and the available data, which included precipitation, temperature, accumulated temperature, the aridity index, the humidity index, the digital elevation model, the surface slope, the soil's bulk density, the soil's organic carbon, the soil's total nitrogen, the soil's clay content, pH, irrigation, tillage methods, and crop species. Therefore, a model training dataset, S, was organized as:

$$S = \{ (X_i, Y_i),\ i = 1, 2, \ldots, N\}$$

where *X* is an M-dimensional vector of the explanatory parameters, and *Y* is the target parameters (RR).

From the training dataset, $n_{tree}$ subset $S_k$ (k = 1, 2, . . . , $n_{tree}$) was randomly selected by using the bootstrap resampling method to generate the regression tree model. For each regression tree, mtry (mtry < M) features were randomly selected at each node, and all split points of these features were traversed for finding the optimal split in minimizing the sum of squares error between the estimated and the real values. For example, consider a split variable, *j*, and split points, and define the pair of half-planes as follows:

$$R_1(j,s) = \{X|X_j \le s\}\ and\ R_2(j,s) = \{X|X_j > s\}$$

Then, the split variable, *j*, and split point, *s*, that solve the following were sought:

$$\min(j,s) \left[ min_{c1} \sum_{yi \in R_1(j,s)} (yi - c1)^2 + min_{c2} \sum_{yi \in R_2(j,s)} (yi - c2)^2 \right]$$

where $c1$ is the average output value for dataset $R_1$, and $c2$ is the average output value for dataset $R_2$.

When finding the optimal split, data were separated into two resulting regions, consequently repeating this splitting process on each of the two sub-nodes, which was stopped if a minimum node (number of observations in a terminal node) was reached.

The ensemble of all the regression trees hi($X$), $i = 1, 2, \ldots, n_\text{tree}$ outputs the final prediction (RF) as follows:

$$f(X) = \frac{1}{n_{tree}} \sum_{i=1}^{n_{tree}} hi(X)$$

The general errors of prediction, based on *OOB*, are calculated as:

$$MSE_{OOB} = \frac{1}{n} \sum_{i=1}^{n} \left(yi - yi^{OOB}\right)^2$$

where *OBB* (out-of-bag data) was approximately 37% of the training data, *S*, which is un-selected in each bootstrap sample, $S_k$.

Secondly, we used the RF model to predict the yields of CS-to-CT and DT-to-CT in Northeast China. In doing this, the basic information, i.e., the support datasets, were imported as the model inputs and the outputs of the RF model, i.e., the predicted RR under CS-to-CT or DT-to-CT. Ultimately, the predicted RR was projected onto a cropland map of Northeast China, with the percent changes [(exp(ln(RR)) $-1$) $\times$ 100%]).

In the present analysis, we used the package "party" in R (version 3.6.1) software to calculate the RF model and variable importance with the functions of "cforest" and "varimp". Based on the minimized OOB mean squared error, the number of trees to grow in each forest was set at 1000, the number of observations at the terminal nodes of the trees was set at 2, and the number of randomly selected features, $m_\text{try}$, for node splitting was set at 3 to create the RF model. In completing these, all calculation processes were run on the super-computer machine from the Beijing Super Cloud Computing Center.

### 3.2.3. Model Accuracy and Validation

To evaluate the model's performance, we randomly choose 70% of the training data as the inputs and another 30% as the validation data. Furthermore, MAE and RMSE measures were calculated to estimate the accuracy of the RF model, as follows:

$$MAE = \frac{\sum_{i=1}^{n} |pi - o|}{n}$$

$$RMSE = \sqrt{\frac{1}{n} \sum_{i=1}^{n} (pi - o)^2}$$

where *MAE* is the mean absolute error, *RMSE* is the root mean square error, $R^2$ is the regression coefficient of determination, $P_i$ is the estimated value, $O_i$ is the real value, and *O* is the average of the real values.

Finally, the linear regression relationship between the estimated values and real values was tested to evaluate the model validation, and the correlation coefficient, *R*, is shown in Figure 7. In general, the performance (see the *MAE*, *RMSE*, and *R* values in the figure) of these two models was good to acceptable in large areas of prediction; however, it was hard to forecast one precise point.

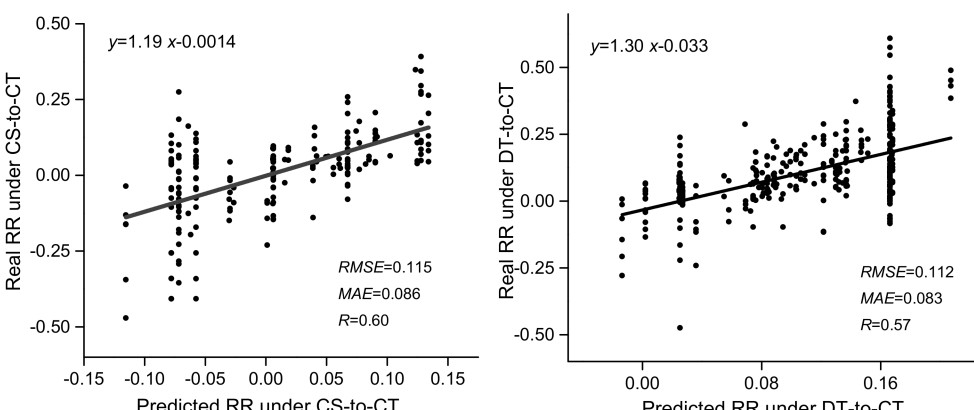

**Figure 7.** The linear relationship between the predicted values and the real values under different tillage.

## 4. Conclusions

A large-scale literature search of scientific journal articles was conducted for conservation tillage, deep tillage, and conventional tillage under Mollisols in Northeast China. Furthermore, we found that conservation tillage had higher soil bulk density, strong soil penetration resistance, higher water content, and lower soil temperature, and was suitable for dry and wind erosion-sensitive regions, i.e., the southwest areas of the Northeast. Conversely, deep tillage had better performance in the middle belt of the Northeast China Plain, which contained a lower soil temperature and humid areas. Finally, we provided an original dataset from papers (dataset 1), and two predicted datasets of crop yield changes (dataset 2 and dataset 3) based on the random forest model.

**Supplementary Materials:** The following supporting information can be downloaded at: https: //www.mdpi.com/article/10.3390/data8010006/s1.

**Author Contributions:** M.U.I. and S.H. proposed the research concept; F.J. and F.D. collected the initial data and wrote the manuscript; Z.C. and G.C. prospered the figures and tables; Y.W. and Y.G. collected the initial data. All authors have read and agreed to the published version of the manuscript.

**Funding:** This research was funded by the Key Research and Development Program of Jiangxi Province (20181BBF68009; 20203BBFL63068).

**Institutional Review Board Statement:** Not applicable.

**Informed Consent Statement:** Not applicable.

**Data Availability Statement:** Data is within the article.

**Conflicts of Interest:** The authors declare no conflict of interest.

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
