# Peer review of "A Large-Scale Dataset of Conservation and Deep Tillage in Mollisols, Northeast Plain, China"

_data, 2022_

Round 1

Reviewer 1 Report

The manuscript describes a large-scale dataset of conservation and deep tillage in Mollisols, Northeast Plain, China.

The text is sometimes hard to read, e.g., ls. 18-19, 36-39, “… is to feed a growing and more demanding world population with degraded soil…” etc; language revision is a pre-requisite for this article to continue. Other points are raised below.

1.           A Conclusions section should be added where the authors can highlight the novelty of their findings.

2.           I believe that, unless prohibited by the Editorial Office, all articles used for data acquisition (61 publications) should be cited, i.e., included in the references section.

3.           Figure 1h: Please, explain why the box of DT is missing.

4.           L.142: “more water contents”: is that true for fig. 1e?

5.           Please, comment on the effect of fertilizers addition.

6.           Please, explain why “… deep tillage was better in the middle belt of Northeast China Plain than in other areas (ls.184-5)” and “conservation tillage (CS) had a better performance in the southwest (l.169)”

7.           Section 3.1.1: Why was not Scopus included?

8.           Ls. 51-2: please, cite the journal article.

Reviewer 2 Report

Some major comments:
1. Introduction
(1) The contribution summary should be listed in Introduction.
(2) The description of the references just comes as a list without providing information about the underlying concepts.

2. Approach
In my opinion, the main weak points of the current submission are its problems with presentation and integration of existing methods. It is not totally clear what the objective is. It is necessary to fully explain the detailed approach to judge its contribution.

3. Evaluation
(1) An explanation of why the authors did these various experiments should be provided.
(2) The author should give more convincing explanations. It would be more convincing to see the performance on more complicated tests.
(3) All experiments should be improved by clarifying the experimental setup.

4. Writing
The consistency and the technical soundness of the paper need to be significantly improved.

Reviewer 3 Report

The paper needs major revisions to be accepted.

Reviewer 4 Report

The manuscript entitled “A Large-Scale Dataset of Conservation and Deep tillage in Mollisols, Northeast Plain, China” contains interesting datasets. It is based on very large review of the literature (61 publications which were selected from more than 300 publications).

The manuscript is quite well prepared however contains some drawbacks.

1) Table 1 is not self-explanatory, it is not clear. Please add more detailed description what the abbreviations of the crops mean and what is presented in graphical form in columns CT and DT. Alle the tables and the figures should be self-explanatory.

2) In the dataset in Excel file some columns have not sufficient description, e.g. lack of units. It should be more specific in what unit is each variable.

3) There is lack information if the all experimental data were conducted in similar way. Please notice that yield can be dependent on type of experiment. At small plot scale yields are usually much higher in comparison to large field experiments. Please explain if such kind of the problem existed in case of the experimental results or not.

4) It would be good if uncertainty layers will be generated since the RMSE is quite high.

Round 2

Reviewer 1 Report

No comments

Reviewer 3 Report

The response to the revision is OK. I recommend accepting the paper.